# A Mott insulator continuously connected to iron pnictide superconductors

Yu Song[1], Zahra Yamani[2], Chongde Cao[1,3], Yu Li[1], Chenglin Zhang[1], Justin S. Chen[1], Qingzhen Huang[4], Hui Wu[4,5], Jing Tao[6], Yimei Zhu[6], Wei Tian[7], Songxue Chi[7], Huibo Cao[7], Yao-Bo Huang[8,9], Marcus Dantz[8], Thorsten Schmitt[8], Rong Yu[10,11,12], Andriy H. Nevidomskyy[1], Emilia Morosan[1], Qimiao Si[1] & Pengcheng Dai[1]

Iron-based superconductivity develops near an antiferromagnetic order and out of a bad-metal normal state, which has been interpreted as originating from a proximate Mott transition. Whether an actual Mott insulator can be realized in the phase diagram of the iron pnictides remains an open question. Here we use transport, transmission electron microscopy, X-ray absorption spectroscopy, resonant inelastic X-ray scattering and neutron scattering to demonstrate that $NaFe_{1-x}Cu_xAs$ near $x \approx 0.5$ exhibits real space Fe and Cu ordering, and are antiferromagnetic insulators with the insulating behaviour persisting above the Néel temperature, indicative of a Mott insulator. On decreasing $x$ from 0.5, the antiferromagnetic-ordered moment continuously decreases, yielding to superconductivity $\sim x = 0.05$. Our discovery of a Mott-insulating state in $NaFe_{1-x}Cu_xAs$ thus makes it the only known Fe-based material, in which superconductivity can be smoothly connected to the Mott-insulating state, highlighting the important role of electron correlations in the high-$T_c$ superconductivity.

[1] Department of Physics and Astronomy and Rice Center for Quantum Materials, Rice University, Houston, Texas 77005, USA. [2] Canadian Neutron Beam Centre, Chalk River Laboratories, Chalk River, Ontario K0J 1J0, Canada. [3] Department of Applied Physics, Northwestern Polytechnical University, Xian 710072, China. [4] NIST Center for Neutron Research, National Institute of Standards and Technology, Gaithersburg, Maryland 20899, USA. [5] Department of Materials Science and Engineering, University of Maryland, College Park, Maryland 20742-2115, USA. [6] Department of Condensed Matter Physics and Materials Science, Brookhaven National National Laboratory, Upton, New York 11973, USA. [7] Quantum Condensed Matter Division, Oak Ridge National Laboratory, Oak Ridge, Tennessee 37831, USA. [8] Paul Scherrer Institute, Swiss Light Source, CH-5232 Villigen PSI, Switzerland. [9] Beijing National Laboratory for Condensed Matter Physics, and Institute of Physics, Chinese Academy of Sciences, Beijing 100190, China. [10] Department of Physics and Beijing Key Laboratory of Opto-electronic Functional Materials & Micro-nano Devices, Renmin University of China, Beijing 100872, China. [11] Department of Physics and Astronomy, Shanghai Jiao Tong University, Shanghai 200240, China. [12] Collaborative Innovation Center of Advanced Microstructures, Nanjing 210093, China. Correspondence and requests for materials should be addressed to Q.S. (email: qmsi@rice.edu) or to P.D. (email: pdai@rice.edu).

At the heart of understanding the physics of the iron-based superconductors is the interplay of superconductivity, magnetism and bad-metal behaviour[1–6]. A key question is whether superconductivity emerges due to strong electronic correlations[6–11] or nested Fermi surfaces[12–14]. Superconductivity occurs in the vicinity of antiferromagnetic (AF) order, both in the iron pnictides and iron chalcogenides[2–5]. In addition, the normal state has a very large room temperature resistivity, which reaches the Ioffe–Mott–Regel limit[2,4,5]. This bad-metal behaviour has been attributed to the proximity to a Mott transition[7–11], with the Coulomb repulsion of the multi-orbital $3d$ electrons of the Fe ions being close to the threshold for electronic localization. In the iron chalcogenide family, several compounds have been found to be AF and insulating with characteristic features of a Mott insulator[15–18]. There is also evidence for an orbital-selective Mott phase in $A_xFe_{2-y}Se_2$ ($A = K$, Rb)[19]. However, in these iron chalcogenide materials, one cannot continuously tune the AF Mott-insulating state into a superconductor. On the other hand, the iron pnictide $NaFe_{1-x}Cu_xAs$ is a plausible candidate for a Mott insulator based on transport and scanning tunnelling microscopy (STM) measurements near $x = 0.3$ (refs 20,21); however, the insulating behaviour may also be induced by Anderson localization via Cu-disorder.

The parent compounds of iron pnictide superconductors such as $AFe_2As_2$ ($A = Ba$, Sr, Ca) and NaFeAs have crystal structures shown in Fig. 1a,b, respectively[2]. They exhibit a tetragonal-to-orthorhombic structural phase transition at temperature $T_s$, followed by a paramagnetic to AF phase transition at $T_N$ ($T_s \geq T_N$) with a collinear magnetic structure, where the spins are aligned antiferromagnetically along the $a$ axis of the orthorhombic lattice (Fig. 1c)[22,23]. When Fe in $AFe_2As_2$ is replaced by Cu to form $ACu_2As_2$ (ref. 24), the Cu atoms have a nonmagnetic $3d^{10}$ electronic configuration with $Cu^{1+}$ and a filled $d$ shell due to the presence of a covalent bond between the As atoms within the same unit cell $[As]^{-3} \equiv [As–As]^{-4}/2$ shown as the shaded As–As bond in Fig. 1a[25–28], as predicted by band structure calculations[29]. Since the crystal structure of $NaFe_{1-x}Cu_xAs$ does not allow a similar covalent bond (Fig. 1b)[20], it would be interesting to explore the magnetic state of $NaFe_{1-x}Cu_xAs$ in the heavily Cu-doped regime. From transport measurements on single crystals of $NaFe_{1-x}Cu_xAs$ with $x \leq 0.3$, it was found that the in-plane resistivity of the system becomes insulating-like for $x \geq 0.11$ (ref. 20). STM showed that local density of states for $NaFe_{1-x}Cu_xAs$ with $x = 0.3$ resembles electron-doped Mott insulators[21], although it is unclear what the undoped Mott-insulating state is or whether it exhibits AF order.

Using transport, neutron scattering, transmission electron microscopy (TEM), X-ray absorption spectroscopy (XAS) and resonant inelastic X-ray scattering (RIXS), we demonstrate that heavily Cu-doped $NaFe_{1-x}Cu_xAs$ exhibits Fe and Cu ordering, and becomes an AF insulator when $x$ approaches 0.5. The insulating behaviour persists into the paramagnetic state, providing strong evidence for a Mott-insulating state. This conclusion is corroborated by our theoretical calculations based on the $U(1)$ slave-spin approach[30]. Our calculations demonstrate enhanced correlations and a Mott localization from the combined effect of a bandwidth reduction, which results from a Cu-site blockage of the kinetic motion of the Fe $3d$ electrons, and a hole doping from $3d^6$ to $3d^5$, both of which are made possible by the fully occupied Cu $3d$ shell[7,9–11,30]. On decreasing $x$ from $x \approx 0.5$, the correlation lengths of Fe and Cu ordering, and magnetic ordering, as well as the ordered magnetic moment continuously decrease, connecting smoothly with the superconducting phase in $NaFe_{1-x}Cu_xAs$ appearing near $x = 0.05$, highlighting the role of electronic correlations in iron pnictides.

## Results

**Resistivity measurements**. Single crystals of $NaFe_{1-x}Cu_xAs$ were prepared using the self-flux method. Transport measurements were carried out using a commercial physical property measurement system with the standard four-probe method (the Methods section). Figure 1e shows temperature dependence of the in-plane resistivity for the $x = 0.016$ sample. In addition to superconductivity at $T_c = 11$ K, the normal state resistivity $\rho$ is $\sim 0.4$ m$\Omega$ cm at room temperature, consistent with previous work[20]. Figure 1f plots the temperature dependence of the resistivity on a log scale for samples with $x = 0.18$, 0.38, 0.44 and 0.48. Resistivity in the $x = 0.18$ sample exhibits insulating-like behaviour consistent with earlier work[20]. For the $x \geq 0.39$ samples, resistivity further increases by an order of magnitude compared to the $x = 0.18$ sample, signifying that electrons become much more localized in $NaFe_{1-x}Cu_xAs$, when $x$ approaches 0.5.

**Neutron scattering results on $NaFe_{0.56}Cu_{0.44}As$**. Unpolarized and polarized elastic neutron scattering were carried out on $NaFe_{1-x}Cu_xAs$ single-crystal samples (the Methods section). For these measurements, we use the orthorhombic unit cell notation suitable for the AF-ordered state of NaFeAs (refs 22,23), and define momentum transfer $\mathbf{Q}$ in three-dimensional reciprocal space in Å$^{-1}$ as $\mathbf{Q} = H\mathbf{a}^\star + K\mathbf{b}^\star + L\mathbf{c}^\star$, where $H$, $K$ and $L$ are Miller indices and $\mathbf{a}^\star = \hat{\mathbf{a}}2\pi/a$, $\mathbf{b}^\star = \hat{\mathbf{b}}2\pi/b$ and $\mathbf{c}^\star = \hat{\mathbf{c}}2\pi/c$ (Fig. 1c). Single crystals are aligned in the $(H, 0, L)$ and $(H, H, L)$ scattering planes in these measurements. In the $(H, 0, L)$ scattering geometry, the collinear magnetic structure in NaFeAs (Fig. 1c) gives magnetic Bragg peaks below $T_N$ at $\mathbf{Q}_{AF} = (H, 0, L)$, where $H = 1, 3, \cdots$ and $L = 0.5, 1.5, \cdots$, positions[22,23].

Figure 2c shows typical elastic scans for $NaFe_{0.56}Cu_{0.44}As$ along the $[H, 0, 0.5]$ direction at $T = 3.6$ and 300 K. Because of twinning in the orthorhombic state, this is equivalent to elastic scans along the $[0, K, 0.5]$ direction. While the scattering has a clear peak centred around $\mathbf{Q} = (1, 0, 0.5)/(0, 1, 0.5)$ at room temperature, the scattering is enhanced markedly on cooling to 3.6 K, suggesting the presence of static AF order. The temperature difference plot between 3.6 and 300 K in Fig. 2d indicates that the low-temperature intensity gain is essentially instrumental resolution limited (horizontal bar). Figure 2e shows the temperature difference along the $[1, 0, L]$/$[0, 1, L]$ direction that apparently is not resolution limited. The temperature dependence of the scattering at $(1, 0, 0.5)$/$(0, 1, 0.5)$ plotted in Fig. 2f reveals clear evidence of magnetic ordering below $T_N \approx 200$ K, suggesting that the small peak observed at room temperature in Fig. 2c occurs in the paramagnetic phase and is thus of structural (super-lattice) origin induced by Cu substitution, since such a peak is forbidden for NaFeAs.

**Fe and Cu ordering in $NaFe_{1-x}Cu_x$ with $x \approx 0.5$**. To conclusively determine the origin of super-lattice peaks and the crystal structure of $NaFe_{0.56}Cu_{0.44}As$, we have carried out high-resolution TEM measurements. The inset in Fig. 2a shows an image of the sample, and the electron diffraction patterns were collected from areas $\sim 100$ nm in diameter within a single domain of the sample (see the arrow in the inset of Fig. 2a). A typical diffraction pattern along the [001] zone axis at room temperature is shown in Fig. 2a. While we see clear super-lattice reflections at $H = 1, 3, \cdots$ positions along the $[H, 0, 0]$ direction,

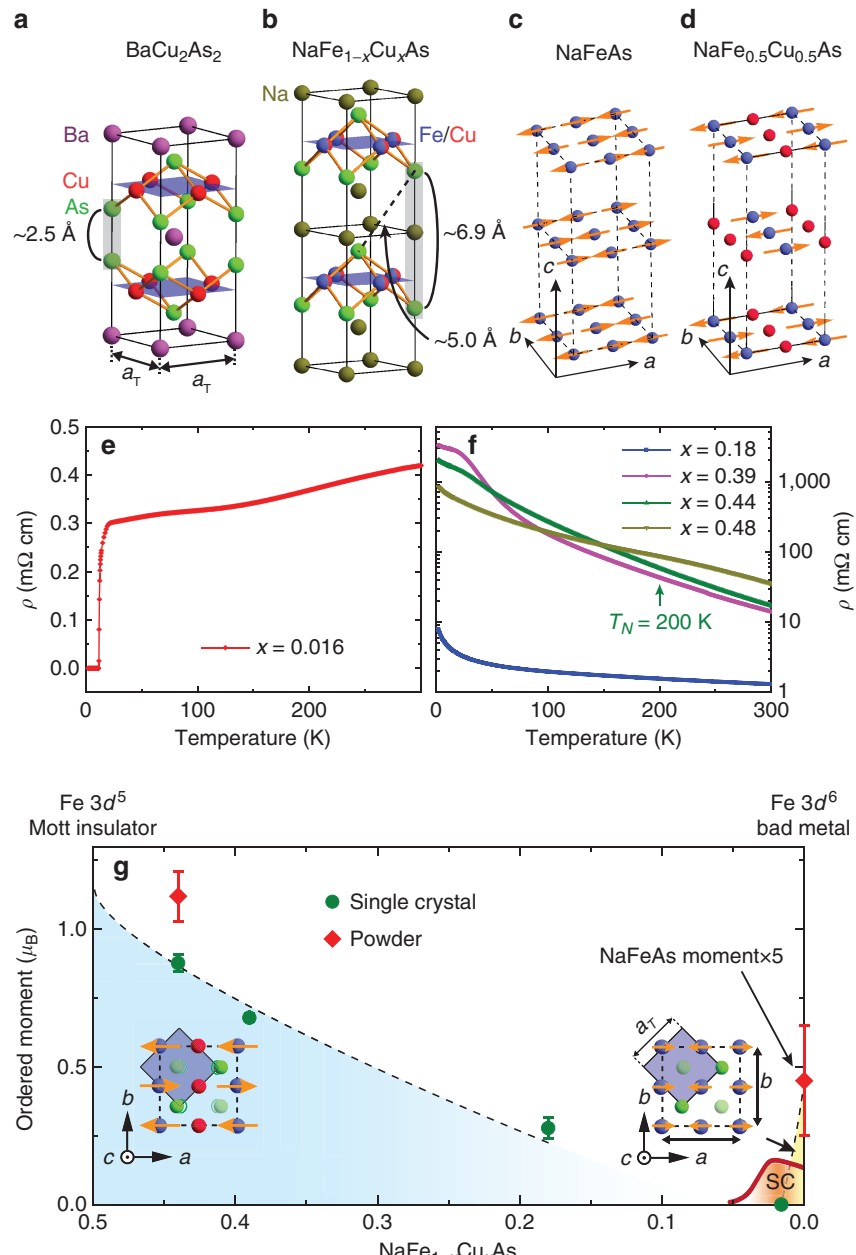

**Figure 1 | Summary of transport and neutron scattering results.** (**a**) The crystal structure of $ACu_2As_2$ in the tetragonal state, where $a_T$ is the in-plane lattice parameter. The As–As covalent bonding distance is $\sim 2.5$ Å. (**b**) The crystal structure of $NaFe_{1-x}Cu_xAs$, where similar As–As covalent bonding is not possible. (**c**) The collinear magnetic structure of NaFeAs, where the AF order and moment direction are along the orthorhombic $a$ axis[22,23]. Only Fe atoms are plotted in the figure for clarity. (**d**) Real space structure and spin arrangements of $NaFe_{0.5}Cu_{0.5}As$ in the AF orthorhombic unit cell similar to NaFeAs. In $NaFe_{0.5}Cu_{0.5}As$, since Fe and Cu form stripes, Na and As also shift slightly from their high-symmetry positions. (**e**) In-plane resistivity for $x = 0.016$ sample, where bulk superconductivity occurs below $T_c = 11$ K (ref. 20). (**f**) Temperature dependence of the in-plane resistivity for $x = 0.18$, 0.39, 0.44 and 0.48 samples. The vertical arrow marks the position of $T_N$ for the $x = 0.44$ sample determined from neutron scattering. (**g**) Evolution of ordered moment with doping in $NaFe_{1-x}Cu_xAs$. The two ordered phases are separated by the superconducting dome marked as SC, with no magnetic order for $x = 0.016$ near optimal superconductivity. The right and left insets show the in-plane magnetic structures of NaFeAs and the new AF insulating phase, respectively. For $x \leq 0.05$, the phase diagram from ref. 20 is plotted. Vertical error bars are from least-square fits (1 s.d.).

they are absent at the $K = 1, 3, \cdots$ positions along the $[0, K, 0]$ direction (Fig. 2b). This means that the crystal structure of $NaFe_{0.56}Cu_{0.44}As$ is orthorhombic and obeys two-fold rotational symmetry. From this information and from intensities of the super-lattice peaks in $(H, 0, L)$ scattering plane, we conclude that Fe and Cu atoms in $NaFe_{0.5}Cu_{0.5}As$ (which is approximated by $NaFe_{0.56}Cu_{0.44}As$) form a real space stripe-like-ordered structure in Fig. 1d, in the space group *Ibam* (Supplementary Note 1;

Supplementary Table 1). This conclusion is further corroborated by single-crystal neutron diffraction measurements (Supplementary Table 2). The Fe–Cu ordering is a structural analogue of the magnetic order in NaFeAs and structural super-lattice peaks occur at the same positions as magnetic peaks in NaFeAs. Fe and Cu ordering in $NaFe_{1-x}Cu_xAs$ affects the intensities of Bragg peaks already present in NaFeAs very little (Supplementary Note 1; Supplementary Fig. 1), and the induced

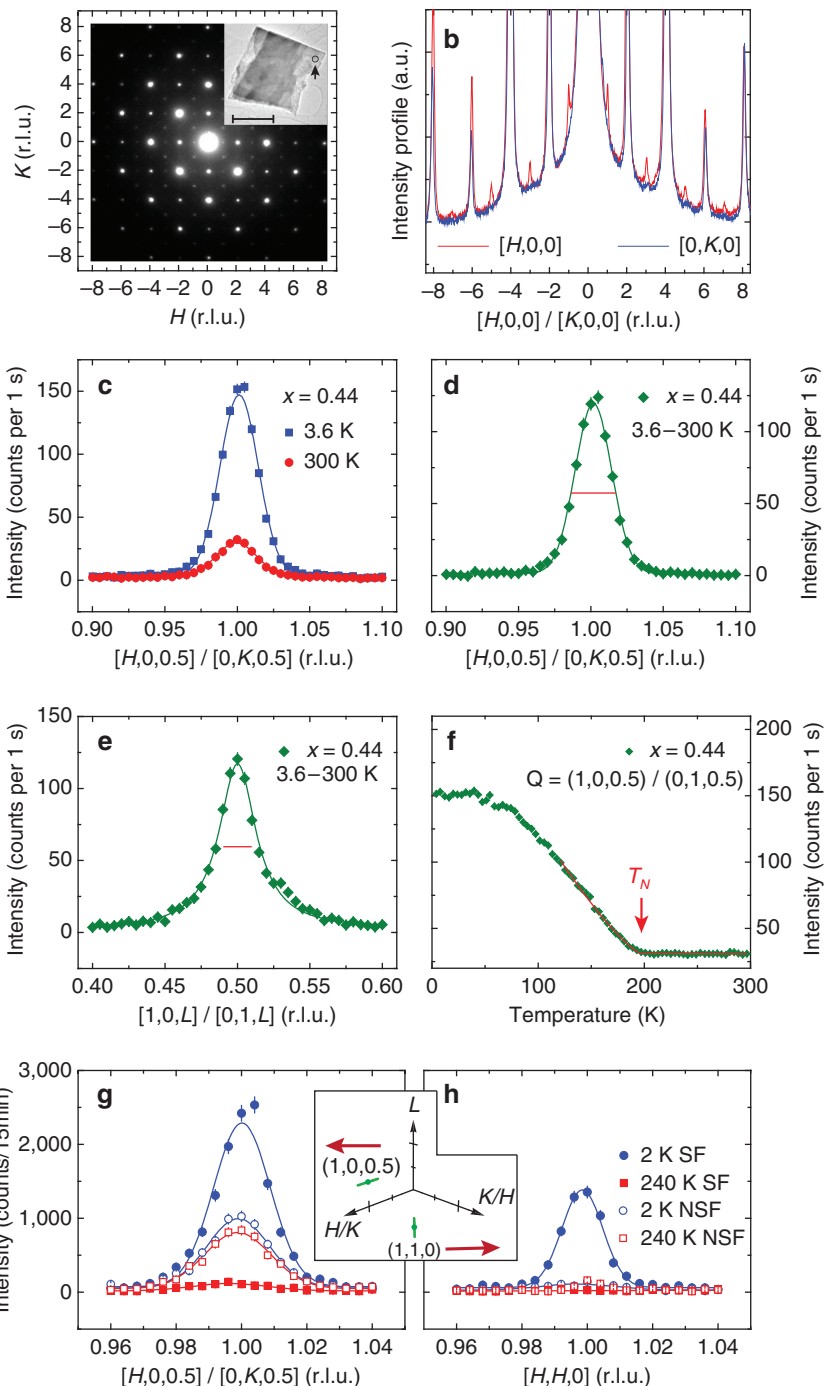

**Figure 2 | TEM and neutron scattering results on the structure and magnetic order for NaFe$_{0.56}$Cu$_{0.44}$As.** (**a**) Using 200 keV incident electrons, an electron diffraction pattern in the [$H$, $K$] plane was obtained from a NaFe$_{0.56}$Cu$_{0.44}$As particle with its TEM real space image shown in the inset. Scale bar, is 1 μm. (**b**) Cuts of **a** along the [$H$, 0] and [0, $K$] directions. (**c**) Unpolarized neutron diffraction scans along the [$H$, 0, 0.5]/[0, $K$, 0.5] direction for the $x = 0.44$ sample at 300 and 3.6 K. The peak at 300 K is a super-lattice peak arising from Fe–Cu ordering. Temperature difference plots between 3.6 and 300 K along the (**d**) [$H$, 0, 0.5]/[0, $K$, 0.5] and (**e**) [1, 0, $L$]/[0, 1, $L$] directions. Solid lines are Gaussian fits and the horizontal bars indicate instrumental resolution. (**f**) Temperature dependence of the scattering at $\mathbf{Q} = (1, 0, 0.5)/(0, 1, 0.5)$ shows $T_N \approx 200$ K. (**g**) Neutron polarization analysis of the magnetic Bragg peak at (1, 0, 0.5)/(0, 1, 0.5). Neutron SF and NSF cross-sections are measured at 2 and 240 K. The peak at 240 K is nonmagnetic nuclear scattering giving rise to NSF scattering. (**h**) Similar scans for the (1, 1, 0) peak. The inset shows positions of these two peaks in reciprocal space. All vertical error bars represent statistical error (1 s.d.).

super lattice are relatively weak. Aside from ordering of Fe and Cu, other aspects of the structure of NaFe$_{1-x}$Cu$_x$As remain identical to that of NaFeAs, corroborated by the similar neutron powder diffraction patterns (Supplementary Note 1; Supplementary Fig. 2; Supplementary Table 3).

**Magnetic structure of NaFe$_{1-x}$Cu$_x$As with $x \approx 0.5$.** Having established the real space Cu and Fe ordering and crystal structure of NaFe$_{0.56}$Cu$_{0.44}$As, it is important to determine the magnetic states of Cu and Fe. Assuming that the covalent As–As bonding is not possible because of the large distance between the

neighbouring As ions (Fig. 1b), As ions in $NaFe_{1-x}Cu_xAs$ can only be in the $As^{3-}$ state. From XAS and RIXS experiments on $NaFe_{0.56}Cu_{0.44}As$ (Supplementary Note 2; Supplementary Fig. 3), we conclude that Cu in $NaFe_{1-x}Cu_xAs$ is in the nonmagnetic $Cu^{1+}$ configuration. As a consequence, Fe should be in the $Fe^{3+}$ $3d^5$ state, since Na can only be in the $Na^{1+}$ state.

Assuming Fe in $NaFe_{0.56}Cu_{0.44}As$ is indeed in the $Fe^{3+}$ $3d^5$ state, we can determine the magnetic structure of the system by systematically measuring magnetic peaks at different wave vectors (Supplementary Note 3; Supplementary Figs 4 and 5) and using neutron polarization analysis (Supplementary Note 3; Supplementary Fig. 6). By polarizing neutrons along the direction of the momentum transfer $\mathbf{Q}$, neutron spin-flip (SF) scattering is sensitive to the magnetic components perpendicular to $\mathbf{Q}$, whereas the non-SF (NSF) scattering probes pure nuclear scattering[31,32]. Figure 2g shows SF and NSF scans along the $[H, 0, 0.5]/[0, K, 0.5]$ direction at $T = 2$ and 240 K. Inspection of the data reveals clear magnetic scattering at 2 K that disappears at 240 K, in addition to the temperature-independent NSF nuclear super-lattice reflection. Figure 2h shows the SF and NSF scans along the $[H, H, 0]$ direction at 2 and 240 K. While the SF scattering shows a clear peak at 2 K that disappears on warming to 240 K, the NSF scattering shows no evidence of the super-lattice peaks. From these results and from systematic determination of magnetic and super-lattice scattering intensity at different wave vectors, we conclude that $NaFe_{0.56}Cu_{0.44}As$ forms a collinear magnetic structure with moments aligned along the $a$ axis as shown in the right inset in Fig. 1g. The ordered magnetic moment is $1.12 \pm 0.09$ $\mu_B$/Fe at 4 K determined from neutron powder diffraction (Supplementary Fig. 2e). For this magnetic structure, magnetic peaks are expected and observed at $(0, 1, 0.5)$ and $(1, 1, 0)$ as shown in Fig. 2g,h (Supplementary Fig. 7). The magnetic peaks at $(0, 1, 0.5)$ overlap with super-lattice peaks at $(1, 0, 0.5)$ due to twinning.

**Doping dependence of magnetic order in $NaFe_{1-x}Cu_xAs$.** To determine the Cu-doping dependence of the $NaFe_{1-x}Cu_xAs$-phase diagram, we carried out additional measurements on single crystals of $NaFe_{1-x}Cu_xAs$ with $x = 0.18$ and 0.39. Figure 3a,b compare the wave vector scans along the $[H, 0, 0.5]/[0, K, 0.5]$ and $[1, 0, L]/[0, 1, L]$ directions for $x = 0.18$, 0.39 and 0.44. In each case, the scattering intensity is normalized to the $(2, 0, 0)$ nuclear Bragg peak. With increasing Cu doping, the scattering profile becomes narrower and stronger, changing from a broad peak indicative of the short-range magnetic order in $x = 0.18$ and 0.39 to an essentially instrument resolution limited peak with long-range magnetic order at $x = 0.44$. Figure 3c shows the doping evolution of the spin–spin correlation length $\xi$ in $NaFe_{1-x}Cu_xAs$, suggesting that the increasing spin correlations in $NaFe_{1-x}Cu_xAs$ are related to the increasing Cu doping and the concomitant increase in correlation length of Fe and Cu ordering (Supplementary Note 3; Supplementary Fig. 5f). The evolution of the temperature dependent magnetic order parameter with Cu doping is shown in Fig. 3d. For $x$ well $< 0.5$, the magnetic transition is gradual and spin-glass-like. However, with $x$ approaching 0.5, the transition at $T_N$ becomes more well defined. In the undoped state, NaFeAs has a small ordered moment of $0.17 \pm 0.03$ $\mu_B$/Fe (ref. 23). On small Cu doping, superconductivity is induced at $x = 0.02$ and the static AF order is suppressed[20]. With further Cu doping, the system becomes an AF insulator, where the ordered moment reaches $\sim 1.1$ $\mu_B$/Fe at $x = 0.44$ (the ordered moment per Fe site was determined in the structure for $NaFe_{0.5}Cu_{0.5}As$ (Fig. 1d)). Since the iron moment in $NaFe_{1-x}Cu_xAs$ increases with increasing Cu doping, our determined moment is therefore a lower bound for the ordered moment for Fe in the ideal $NaFe_{0.5}Cu_{0.5}As$, which for Fe $3d^5$ can be in either $S = 3/2$ or $S = 5/2$ spin state.

## Discussion

The behaviour in $NaFe_{1-x}Cu_xAs$ is entirely different from the bipartite magnetic parent phases seen in the iron oxypnictide superconductor $LaFeAsO_{1-x}H_x$, where magnetic parent phases on both sides of the superconducting dome are metallic antiferromagnets[33]. In Cu-doped $Fe_{1-x}Cu_xSe$, a metal–insulator transition has been observed $\sim 4\%$ Cu doping, and a localized moment with spin glass behaviour is found near $x \approx 0.12$ (refs 34,35). Although the density functional theory (DFT) calculations suggest that Cu occurs in the $3d^{10}$ configuration and the metal–insulator transition is by a disorder induced Anderson localization in $Fe_{1-x}Cu_xSe$ (ref. 36), the 20–30% solubility limit of the system[34] means it is unclear whether $Fe_{1-x}Cu_xSe$ is also an AF insulator at $x \approx 0.5$.

It is useful to compare our experimental findings with the results of the band structure calculations based on DFT (the Methods section; Supplementary Note 4; Supplementary Figs 8–10). Both the DFT and DFT $+ U$ calculations clearly predict the paramagnetic phase of $NaFe_{0.5}Cu_{0.5}As$ to be a metal (Supplementary Fig. 9). By contrast, our measurements have shown that the insulating behaviour of the resistivity persists above the Néel temperature (Fig. 1f), implying that $NaFe_{0.5}Cu_{0.5}As$ is a Mott insulator. This is also consistent with STM measurements on lower Cu-doping $NaFe_{1-x}Cu_xAs$, where the overall line shape of the electronic spectrum at $x = 0.3$ is similar to those of lightly electron-doped copper oxides close to the parent Mott insulator[21].

To understand the origin of the Mott-insulating behaviour, we address how Cu doping affects the strength of electron correlations (the Methods section; Supplementary Note 5; Supplementary Figs 11 and 12). Our starting point is that the local (ionization) potential difference between the Fe and Cu ions, as illustrated in Fig. 4a and described by the energy shift $\Delta$ in Fig. 4b, will suppress the hopping integral between Fe and Cu sites. This causes a reduction in the kinetic energy or, equivalently, the electron bandwidth. The ratio of the on-site Coulomb repulsion $U$ (including the Hund's coupling $J_H$) relative to the bandwidth will effectively increase, even if the raw values of $U$ and $J_H$ remain the same as in pure NaFeAs. This would enhance the tendency of electron localization even without considering the disorder effects with increasing $x$ (refs 37,38).

The case of $x = 0.5$ allows a detailed theoretical analysis. Here, the Cu and Fe ions in $NaFe_{1-x}Cu_xAs$ have real space ordering, as discussed above. We study the metal–insulator transition in multi-orbital Hubbard models for $NaFe_{0.5}Cu_{0.5}As$ via the $U(1)$ slave-spin mean-field theory[30]. We use the tight-binding parameters for NaFeAs, and consider the limit of a large local potential difference between Cu and Fe, which transfers charge from Fe to Cu to $Cu^{1+}$ ($n = 10$). The fully occupied Cu $3d$ shell makes the Cu ions essentially as vacancies. Reduction of the Fe $3d$ electron bandwidth is caused by this kinetic blocking effect (Supplementary Fig. 11). The resulting ground-state phase diagram is shown in Fig. 4c. For realistic parameters, illustrated by the shaded region, a Mott localization takes place. Our understanding is in line with the general theoretical identification of a Mott-insulating phase in an overall phase diagram[30], which takes into account a kinetic energy reduction induced increase of the effective interactions and a decrease of the $3d$ electron-filling from 6 per $Fe^{2+}$ to 5 per $Fe^{3+}$ (refs 7,9–11,30).

Our work uncovers a Mott insulator, $NaFe_{0.5}Cu_{0.5}As$, and provides its understanding in an overall phase diagram of both bandwidth and electron-filling controls[30]. Our results suggest that

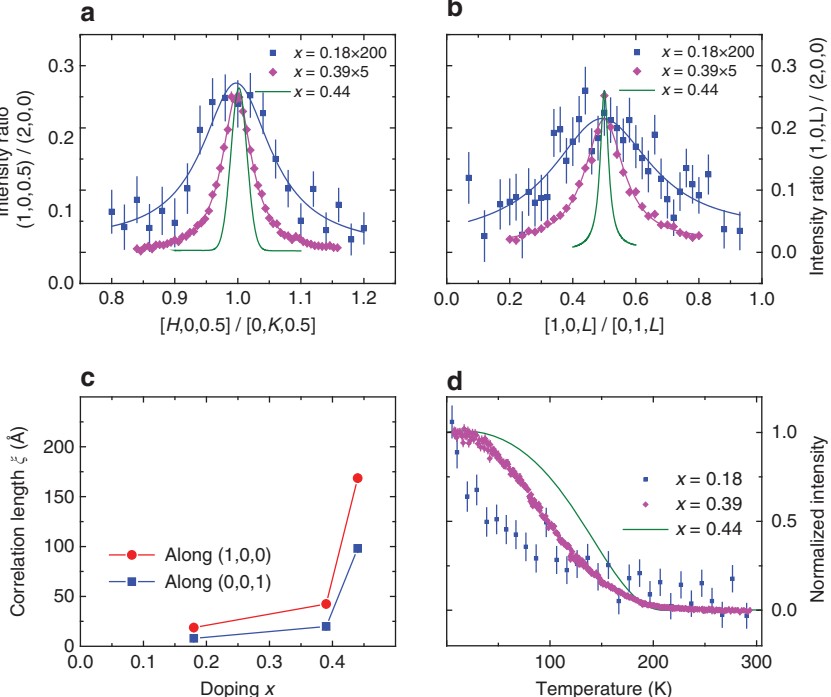

**Figure 3 | Cu-doping evolution of the magnetic order in NaFe$_{1-x}$Cu$_x$As.** (**a**) Comparison of wave vector scans along the [$H$, 0, 0.5]/[0, $K$, 0.5] direction for NaFe$_{1-x}$Cu$_x$As single crystals with $x = 0.18$, 0.39 and 0.44. The data are normalized to the (2, 0, 0) nuclear Bragg peak. Note that intensity in the $x = 0.18$ and 0.39 samples are multiplied by 200 and 5 times, respectively. (**b**) Similar scans along the [1, 0, $L$]/[0, 1, $L$] direction. (**c**) Cu-doping evolution of the spin–spin correlation length. The correlation length along (1, 0, 0) for the $x = 0.44$ sample is resolution limited. (**d**) Cu-doping evolution of the magnetic order parameter for NaFe$_{1-x}$Cu$_x$As. All error bars represent statistical error (1 s.d.).

the electron correlations of the iron pnictides, while weaker than those of the iron chalcogenides, are sufficiently strong to place these materials in proximity to the Mott localization. This finding makes it natural that the electron correlations and the associated bad-metal behaviour and magnetism induce a similarly high superconducting transition temperature in the iron pnictides as in the iron chalcogenide family. Such a commonality, in spite of the very different chemical composition and electronic structure of these two broad classes of materials, introduces considerable simplicity in the quest for a unified framework of the iron-based superconductivity. More generally, the proximity to the Mott transition links the superconductivity of the iron pnictides to that arising in the copper oxides, organic charge-transfer salts[39] and alkali-doped fullerides[40], and suggests that the same framework may apply to all these strongly correlated electronic systems.

Note added after submission: very recently, angle-resolved photoemission spectroscopy measurements on NaFe$_{0.56}$Cu$_{0.44}$As samples also confirmed its Mott-insulating nature[41].

## Methods

**Sample preparation and experimental details.** NaFe$_{1-x}$Cu$_x$As single crystals were grown by the self-flux method using the same growth procedure as for NaFe$_{1-x}$Co$_x$As described in earlier work[42]. The Cu-doping levels used in this paper were determined by inductively coupled plasma atomic-emission spectroscopy. Samples with nominal Cu concentrations of $x = 2$, 20, 50, 75 and 90% were prepared, resulting in actual Cu concentrations of $x = 1.6$, 18.4(0.4), 38.9(2.8), 44.2(1.7) and 48.4(3.4)%. For each nominal doping except $x = 2$%, five to six samples were measured and the s.d. in these measurements are taken as the uncertainty of the actual concentrations. For simplicity, the actual concentrations are noted as $x = 1.6$, 18, 39, 44 and 48% in the rest of the paper. This suggests that the solubility limit of Cu in NaFe$_{1-x}$Cu$_x$As single crystals by our growth method is $\approx 50$%.

For resistivity measurements, samples were mounted onto a resistivity puck inside an Ar filled glove box by the four-probe method and covered in Apiezon N grease. The prepared puck is then transferred in an Ar-sealed bottle and is only briefly exposed in air, while being loaded into a physical property measurement

system for measurements. After measurements, no visual deterioration of the samples were seen under a microscope.

For single-crystal elastic neutron scattering measurements, samples were covered with a hydrogen-free glue and then stored in a vacuum bottle at all times except during sample loading before the neutron scattering experiment. While our largest crystals grown are up to 2 g, we typically used thin plate-like samples $\sim 5 \times 5$ mm$^2$ in size with mass $\approx 0.2$ g for elastic neutron scattering measurements. Measurement of the sample with $x = 0.18$ was carried out on the N5 triple-axis spectrometer at the Canadian Neutron Beam Center, Chalk River Laboratories. Pyrolitic graphite (PG) monochromator and analyser ($E_i = 14.56$ meV) were used with none-36'-sample-33'-144' collimation set-up. A PG filter was placed after the sample to eliminate contamination from higher-order neutron wavelengths. The experiment on $x = 0.39$ and 0.44 samples were done on the HB-3 triple-axis spectrometer and the HB-1A fixed-incident-energy triple-axis spectrometer, respectively, at the High Flux Isotope Reactor, Oak Ridge National Laboratory. On HB-3, a PG monochromator ($E_i = 14.7$ meV), analyser and two PG filters (one before and the other after the sample) were used with collimation set-up 48'-60'-sample-80'-120'. HB-1A uses two PG monochromators ($E_i = 14.6$ meV) and two PG filters (mounted before and after second monochromator), resulting in negligible higher-order wavelengths neutrons in the incident beam. A PG analyser is placed after the sample. The collimation used was 48'-48'-sample-40'-68'. In all cases above, the samples were aligned in [$H$, 0, $L$] scattering plane.

The polarized single-crystal neutron scattering measurements were carried out on the C5 polarized beam triple-axis spectrometer at Canadian Neutron Beam Center, Chalk River Laboratories. The neutron beams were polarized with Heusler (1, 1, 1) crystals with a vertically focusing monochromator and a flat analyser ($E_f = 13.70$ meV). A PG filter was placed after the sample and none-48'-51'-144' collimation was used. Neutron polarization was maintained by using permanent magnet guide fields. Mezei flippers were placed before the sample to allow the measurement of neutron SF and NSF scattering cross-sections. A 5-coil Helmholtz assembly was used to control the neutron spin orientation at the sample position by producing a magnetic field of the order of 10 G. The orientation of the magnetic field at the sample position was automatically adjusted to allow the measurements to be performed for the neutron spin to be parallel or perpendicular to the momentum transfer. The flipping ratio, defined as the ratio of nuclear Bragg peak intensities in NSF and SF channels, was measured to be $\sim 10:1$ for various field configurations. The sample was studied in both [$H$, 0, $L$] and [$H$, $H$, $L$] scattering planes.

The XAS and RIXS measurements on the $x = 0.44$ samples were carried out at the Advanced Resonant Spectroscopy beamline of the Swiss Light Source, Paul Scherrer Institut, Switzerland[43]. Samples were cleaved *in situ* and measured in a

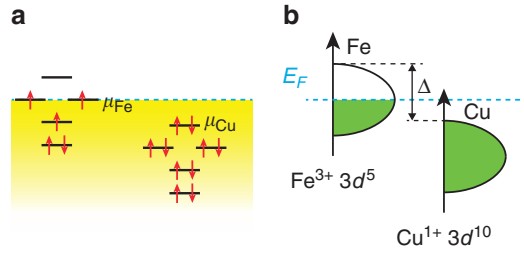

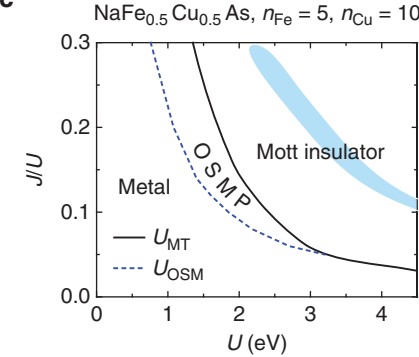

**Figure 4 | Schematic 3d electronic states and parameter regimes of a Mott insulator in the theoretical phase diagram for NaFe_0.5Cu_0.5As.** Schematic of the electronic states of a Cu ion alongside those of an Fe ion, shown in terms of the atomic levels (**a**) and the electronic density of states (**b**). The relative potential difference associated with the two ions is specified in terms of an energy shift, $\Delta$, as illustrated in **b**. (**c**) The ground-state phase diagram of a multi-orbital Hubbard model for NaFe_0.5Cu_0.5As. $U_{OSM}$ and $U_{MT}$ refer to the critical $U$ values for an orbital-selective Mott (OSM) transition and a Mott transition (MT), respectively into an orbital-selective Mott phase (OSMP) and a Mott insulator. The shaded region shows the physical parameter regime as determined by comparing the theoretically calculated bandwidth renormalization factors of the three $t_{2g}$ orbitals with those determined by the angle-resolved photoemission spectroscopy (ARPES) results (Supplementary Note 5).

vacuum better than $1 \times 10^{-10}$ mbar. X-ray absorption was measured in total electron yield mode by recording the drain current from the samples and in total fluorescence yield with a photodiode. Linearly polarized X-rays with $E = 931.8$ eV resonant at the Cu $L_3$ edge were used for the RIXS measurements. The total momentum transfer was kept constant, but the component of momentum transfer in the $ab$ plane has been varied.

For neutron powder diffraction measurements, the samples were ground from 2 g ($x = 0.016$, 0.18, 0.39 and 0.44) of single crystals and sealed in vanadium sample cans inside a He-filled glove box. Neutron powder diffraction measurements were carried out at room temperature (300 K) on the BT1 high-resolution powder diffractometer at NIST Center for Neutron Research. The Ge(3, 1, 1) monochromator was used to yield the highest neutron intensity and best resolution at low scattering angles. For the $x = 0.44$ sample, measurement at 4 K using a Cu(3, 1, 1) monochromator was also carried out.

For the single-crystal neutron diffraction measurement, a sample with $x = 0.44$ (25 mg) was used. The experiment was carried out at the four-circle diffractometer HB-3A at the High Flux Isotope Reactor, Oak Ridge National Laboratory. The data was measured at 250 K with neutron wavelength of 1.003 Å from a bent Si(3, 3, 1) monochromator using an Anger camera detector. Bragg peaks associated with the NaFeAs structure were measured with 1 s per point by carrying out rocking scans. Super-lattice peaks were measured with 10 min per point at each position.

**DFT-based electronic structure calculations.** To gain insight into the insulating nature of Cu-doped NaFeAs, we have performed a series of DFT-based electronic structure calculations. The electronic band structure calculations have been performed using a full-potential linear augmented plane-wave method as implemented in the WIEN2K package[44]. The exchange-correlations function was taken within the generalized gradient approximation (GGA) in the parameterization of Perdew, Burke and Ernzerhof[45]. For the atomic spheres, the muffin-tin radii ($R_{MT}$) were chosen to be 2.5 Bohr for Na sites, 2.38 Bohr for Fe and Cu, and 2.26 Bohr for As sites, respectively. The number of plane waves was limited by a cutoff parameter ($R_{MT} \times K_{max}$) = 7.5. To account for the on-site correlations in the form of the Hubbard parameter $U$, the GGA + $U$ approach in a Hartree–Fock-like scheme was

used[46,47], with the value $U = 3.15$ eV as calculated in LiFeAs using the constrained random-phase approximation (RPA) approach[48].

While the neutron diffraction studies were performed on the $x = 0.44$ sample, it is much easier theoretically to study the 50% Cu-doped NaFe_0.5Cu_0.5As. To model the crystal structure, the experimentally determined lattice parameters and atomic coordinates of NaFe_0.56Cu_0.44As at 4 K were used (Supplementary Table 3). The difference between the structure at $x = 0.44$ and 0.5 was deemed inessential in view of the weak dependence of the lattice parameters on Cu concentration $x$ (Supplementary Table 3). While the experimental structure can be fitted with a symmetry group $P4/nmm$ with disordered Cu/Fe, the high-resolution TEM measurements found that the super-lattice peaks indicate a stripe-like-ordered pattern (Fig. 1d) consistent with the space group $Ibam$. We have therefore used the $Ibam$ space group, with four formula units per unit cell, to perform the $ab\ initio$ calculations. Interestingly, the results of the calculations turn out to be insensitive to the details of the atomic arrangement: for instance, arranging Cu/Fe atoms in the checkerboard pattern results in a very similar density of states as that obtained for the stripe-like atomic configuration. We also note that the adopted ordered arrangement of Fe/Cu sites neglects the effects of disorder and Anderson localization, which may be important at low doping levels $x < 0.5$. Those effects are however beyond the grasp of the first principles DFT calculations.

**$U(1)$ slave-spin mean-field theory.** To take into account the correlation effects beyond the Hatree–Fock level, we further study the metal–insulator transition in multi-orbital Hubbard models for both NaFeAs and NaFe_0.5Cu_0.5As via the $U(1)$ slave-spin mean-field theory[49]. The model Hamiltonian contains a tight-binding part and a local interaction part. Detailed description of the model is given in ref. 49. The tight-binding parameters for NaFeAs are taken from ref. 50, and the electron density is fixed to $n = 6$ per Fe. As for NaFe_0.5Cu_0.5As, the very large local potential difference between Cu and Fe, as estimated from our and previous[37] DFT calculations, leads to charge transferring from Fe to Cu. As a result, Fe is effectively hole doped when the Cu-doping concentration is close to 0.5. This picture is supported by our experimental results which suggest that the Cu is in a $3d^{10}$ configuration with $n = 10$, and Fe configuration is $3d^5$ with $n = 5$. As a zeroth-order approximation to the fully occupied Cu 3d shell, we treat Cu ions as vacancies. They organize themselves in a columnar-order pattern in NaFe_0.5Cu_0.5As, as shown in Fig. 1d.

**Data availability.** The data that support the findings of this study are available from the corresponding authors on request.

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

## Acknowledgements

We thank X.H. Chen, B.J. Campbell and Lijun Wu for helpful discussions, Leland Harriger, Scott Carr, Weiyi Wang and Binod K. Rai for assisting with some experiments. The single-crystal growth and neutron scattering work at Rice is supported by the U.S. DOE, BES under contract no. DE-SC0012311 (P.D.). A part of the material synthesis work at Rice is supported by the Robert A. Welch Foundation Grant No. C-1839 (P.D.). The theoretical work at Rice was in part supported by the Robert A. Welch Foundation Grant No. C-1818 (A.H.N.), C-1411 (Q.S.), by U.S. NSF grants DMR-1350237 (A.H.N.) and DMR-1611392 (Q.S.), and by the Alexander von Humboldt Foundation (Q.S.). E.M. and J.C. acknowledge support from the DOD PECASE. The electron microscopy study at Brookhaven National Laboratory was supported by the U.S. DOE, BES, by the Materials Sciences and Engineering Division under Contract No. DE-AC02-98CH10886. The use of ORNL's High Flux Isotope Reactor was sponsored by the Scientific User Facilities Division, Office of BES, U.S. DOE. XAS and RIXS experiments have been performed at the Advanced Resonant Spectroscopy beamline of the Swiss Light Source at the Paul Scherrer Institute. T.S and M.D. acknowledge funding through the Swiss National Science Foundation within the D-A-CH programme (SNSF Research Grant 200021L 141325). R.Y. acknowledges the support from the National Science Foundation of China Grant number 11374361, and the Fundamental Research Funds for the Central Universities and the Research Funds of Remnin University of China Grant number 14XNLF08. C.C. acknowledges the support from the National Natural Science Foundation of China Grant No. 51471135, the National Key Research and Development Program of China Grant No. 2016YFB1100101 and Shaanxi International Cooperation Program.

## Author contributions

Most of the single-crystal growth and neutron scattering experiments were carried out by Y.S. with assistance from C.C., Y.L., C.Z., Q.H., H.W., W.T., S.C. and H.C. The polarized neutron scattering experiments were performed by Z.Y. TEM measurements were carried out by J.T. and Y.Z. Transport and inductively coupled plasma measurements were carried out by C.C. and J.S.C., under the supervision of E.M. The XAS and RIXS experiments were performed by Y.-B.H., M.D. and T.S. The theoretical work was done by R.Y., A.H.N and Q.S. P.D. provided the overall lead of the project. The paper was written by P.D., Y.S., A.H.N., R.Y. and Q.S. All authors provided comments.

## Additional information

**Competing financial interests:** The authors declare no competing financial interests.

**Publisher's note**: 

