## [Peer Review File · Nature Communications]

Reviewers' comments:

Reviewer #1 (Remarks to the Author):

The authors have submitted a new version of the manuscript which address the criticisms raised by myself and the other Referees. The authors in particular have addressed my main points (1) and (2) of the previous report.

The new version of the manuscript is certainly much clearer and several potentially misleading aspects have been eliminated. Also the discussion about the theoretical literature is more faithful and complete.

In particular the authors acknowledge the ambiguity of the original formulation, where the Mott insulator studied in this manuscript was claimed to be "near" the superconducting state. The authors now clearly state that the key point of the manuscript is to report a smooth connection between the superconductor and the Mott insulator which is realized by substituting Fe with Cu.

I am perfectly fine with this clarification and I agree with the authors that this is the first case where a Mott state is continuously connected with a superconducting state in the iron-superconductor family (although two places can be smoothly connected, yet very far). I am not equally sure that the new version of the title is actually much better than than the original one. In my mind, the expression "entwined" is quite evocative, but it is does not convey a clear physical message. I would suggest the authors to use the expression "smoothly connected" also in the title.

Turning from semantics to physics, I think that the revised manuscript presents more correctly what happens when Fe is replaced by Cu. Based on the experimental information that Cu is in a d10 configuration for $x=0.5$ and on the theoretical calculations for the same endpoint compound, it appears clear that Cu-doping leads to at least two main effects which favor a Mott transition: (1) a change of the effective charge on Fe orbitals from 6 to 5 and (2) a reduction of kinetic energy due to the vacancies. While the two points are reasonably well stressed in the conclusions, in the introductory part the authors still omit the important fact that the iron sites are effectively hole doped (Sentence: "which show that a Cu-site blockage of the kinetic motion of the Fe 3d electrons reduces the electron bandwidth and thus enhances the effect of electron correlations, pushing the system towards Mott localization"). The doping effect should be clearly mentioned also in this section, also citing the papers that put emphasis on the role of the d5 configuration, namely Refs. 10 and 11 and to a lesser extent Ref. 9.

Once the readership is clearly informed that these are the two main effects which both contribute to drive the Mott transition, it can decide if one of the two effects is more important or if they are both necessary. Of course the authors can add their arguments for one of the other mechanism, but this should be separated from the presentation of the actual results.

In this way it is also easier to rationalize in which sense we can consider the $x=0.5$ material close (or far) to the superconductor with $x=0$. If we refer to a generic diagram such as Fig. 7 of Ref. 37, NaFeAs is in the blue region at $N=6$. Cu substitution drives us to some point on the $N=5$ axis with a larger

value of U/W , deeply in the Mott state.

I also find quite unfair to quote Ref. 37 for the “general theoretical investigation of a Mott insulating state in an overall phase diagram”. The diagram of Ref. 37 is a schematic summary of many previous investigations including very basic physical facts and less obvious aspects related to the Hund’s coupling. I think that the above sentence should quote all the above theoretical works where the possible role of both the $n=6$ and $n=5$ Mott transitions has been discussed, including 9-11.

I am satisfied by the other replies by the authors.

Finally, a very minor point. When the authors conclude that the role of strong correlations can be unifying framework of several superconductors, they might add also the alkali-doped fullerides, where the role of strong correlations seems nowadays established [See for example, the recent paper by Nomura et al. *Science Advances* 1, e1500568 (2015)]

In conclusion, I think that the present manuscript can be published in *Nature Communications* once the authors take into account my last comments.

Reviewer #2 (Remarks to the Author):

I have carefully read the revised manuscript and the authors’ response. I appreciate the authors’ performing new x-ray experiments to determine the valence state of Cu. I agree with the authors’ assessment that their set of experimental measurements provides strong evidence for a Mott insulating state for samples with concentrations near $x=0.5$.

The main issues, however, are about the novelty of this finding and its impact to the understanding of superconductivity in iron-based compounds. On the latter issue, which in my view is the most critical one, the several changes made by the authors are not only unconvincing, but they seem to actually support the opposite point of view, namely that this Mott insulating phase seems nearly inconsequential to the superconducting state. The authors now claim that the Mott insulator near $x=0.5$ is not “near”, but rather “entwined” and “smoothly connected” with the superconducting state at $x=0.016$. To support this claim, they now plot in Fig. 1g the “ordered magnetic moment” as function of x , instead of the Neel transition temperature (which is restricted to the region near $x=0.5$). The first problem is that, since there are no data points for concentrations below $x=0.18$, I do not understand how the authors can justify that the “ordered magnetic moment” line shown in Fig. 1g ends precisely at the end of the superconducting dome at much smaller doping levels, $x=0.04$. The second – and most important – issue is that this plot contradicts Fig. 3c: as the authors state in the main text, there is no long-range magnetic order for $x<0.44$, as shown by both resistivity and neutron scattering data. Therefore, there is no meaning in assigning an “ordered magnetic moment” in that region of the phase diagram, where at best one has only short-ranged fluctuations with correlation lengths of a few lattice parameters, as shown in Fig. 3c. Finally, as the authors discuss in the paper, the Mott insulating state seems to be favored by the specific Cu concentration of $x=0.5$, which causes real space Fe/Cu order and changes the valence of Fe to $3d^5$. In this case, it is very hard to conceive a scenario in which this peculiar Mott state is “smoothly connected” to the $x=0$ compound.

On the issue of novelty, although it is true that the STM studies in Ref. [16] could not unambiguously determine the Mott insulating nature of this phase, they did provide solid indirect evidence in favor of this state. Of course, the current work goes well beyond Ref. [16] by extending the phase diagram to $x=0.44$ and by combining a series of thermodynamic measurements that exclude other possible sources for the insulating state. Nevertheless, in my opinion, this does not provide a major

advancement, in view of the issues raised above. With appropriate and substantial revisions, the paper might be suitable for publication in a more specialized journal.

Reviewer #3 (Remarks to the Author):

I have read the revised version of the manuscript and the authors' rebuttal. My summary of the key results and other comments on the quality of the data, presentation and appropriateness of the references remain unchanged. The authors have made a serious effort to address the referees' concerns, including performing further experiments. In particular the authors show convincingly by single crystal neutron diffraction that a substantially fraction of the sample has Fe/Cu order, which strengthens the conclusion that the 50% doped sample is a Mott insulator. In my opinion the manuscript is now suitable for publication in Nature Communications.

Reviewer #4 (Remarks to the Author):

In this manuscript, the authors successfully observe an AF insulating state in heavily Cu-doped NaFe_{1-x}Cu_xAs material. Moreover, the authors claim that the possibility of insulating state due to Anderson localization could be excluded due to the confirmed ordering arrangement of Cu and Fe atoms in the FeAs plane. From additional XAS and RIXS experiments, the authors also conclusively confirm the 3d¹⁰ electronic configuration of Cu in x=0.50 material, which strongly support their previous assertion of 3d⁵ Fe configuration is in fact correct. Based on these facts, I think that the conclusion on Mott insulating phase in x=0.50 material is solid. Based on this finding, the authors also obtained a new phase diagram for Cu-doped NaFeAs as shown in Fig.1(g) which suggests a close connection between Mott physics and superconductivity in this material. The main controversial point between the authors and other referees for this issue is whether the Mott insulator is close enough to NaFeAs based superconductors to be relevant to superconductivity. In my opinion, Mott physics is unambiguously important for superconductivity in iron-based superconductor. The key point is to clarify how Mott physics affects superconductivity here. In present work, the authors compare this material to electron-doped cuprates. Based on the multi-orbital theoretical consideration, the average electron doping level on each orbital here is believed to be less than 20%, comparable with electron-doped cuprates. I think this consideration is reasonable. Moreover, the recent ARPES experiment confirmed the Mott insulating phase and proves that the conduction band is of Fe 3d-like character for heavily Cu-doped NaFe_{1-x}Cu_xAs (arXiv:1607.00770). Therefore, the main conclusion in present work is acceptable. In addition, the authors have made a very good reply for all comments raised by all referees. I think the present manuscript is worth to be published in Nature Communications.

REVIEWERS' COMMENTS:

Reviewer #1 (Remarks to the Author):

The authors have satisfactorily replied to the remaining points that I raised in my report. In particular

1) I appreciate the change in the title, which I think will be beneficial for the impact of the paper.

2) I think that the discussion of the previous literature about the role of Mott physics in iron-based superconductors has been improved.

3) I agree with the authors that the experimental reference on the role of Mott physics in Cs₃C₆₀ is more appropriate than the theory paper I mentioned

Incidentally, I think the authors replied satisfactorily also to the other Referee.

The revised manuscript can be published in Nature Communications in the present form.

Response to referee 1

“The authors have submitted a new version of the manuscript which address the criticisms raised by myself and the other Referees. The authors in particular have addressed my main points (1) and (2) of the previous report.

The new version of the manuscript is certainly much clearer and several potentially misleading aspects have been eliminated. Also the discussion about the theoretical literature is more faithful and complete.

In particular the authors acknowledge the ambiguity of the original formulation, where the Mott insulator studied in this manuscript was claimed to be “near” the superconducting state. The authors

now clearly state that the key point of the manuscript is to report a smooth connection between the superconductor and the Mott insulator which is realized by substituting Fe with Cu.”

We appreciate these positive comments from the referee.

“I am perfectly fine with this clarification and I agree with the authors that this is the first case where a Mott state is continuously connected with a superconducting state in the iron-superconductor family (although two places can be smoothly connected, yet very far). I am not equally sure that the new version of the title is actually much better than than the original one. In my mind, the expression “entwined” is quite evocative, but it does not convey a clear physical message. I would suggest the authors to use the expression “smoothly connected” also in the title.”

We have followed the suggestion of the referee and replaced the term ‘entwined with’ by ‘continuously connected to’ in the title of our manuscript.

“Turning from semantics to physics, I think that the revised manuscript presents more correctly what happens when Fe is replaced by Cu. Based on the experimental information that Cu is in a $d10$ configuration for $x=0.5$ and on the theoretical calculations for the same endpoint compound, it appears clear that Cu-doping leads to at least two main effects which favor a Mott transition: (1) a change of the effective charge on Fe orbitals from 6 to 5 and (2) a reduction of kinetic energy due to the vacancies. While the two points are reasonably well stressed in the conclusions, in the introductory part the authors still omit the important fact that the iron sites are effectively hole doped (Sentence: “which show that a Cu-site blockage of the kinetic motion of the Fe 3d electrons reduces the electron bandwidth and thus enhances the effect of electron correlations, pushing the system towards Mott localization”). The doping effect should be clearly mentioned also in this section, also citing the papers that put emphasis on the role of the $d5$ configuration, namely Refs. 10 and 11 and to a lesser extent Ref. 9.

Once the readership is clearly informed that these are the two main effects which both contribute to drive the Mott transition, it can decide if one of the two effects is more important or if they are both necessary. Of course the authors can add their arguments for one of the other mechanism, but this should be separated from the presentation of the actual results.”

In this way it is also easier to rationalize in which sense we can consider the $x=0.5$ material close (or far) to the superconductor with $x=0$. If we refer to a generic diagram such as Fig. 7 of Ref. 37, NaFeAs is in the blue region at $N=6$. Cu substitution drives us to some point on the $N=5$ axis with a larger value of U/W , deeply in the Mott state.”

We thank the referee for these additional comments. In fact, the substance of these remarks is the same as what we said in our previous reply, which we repeat here for convenience: “Having said the above, we wish to clarify the theoretical meaning of the Mott insulating phase in the overall phase diagram. It is well known that proximity to a Mott insulator can be realized in two kinds of ways: by tuning kinetic energy (in the sense that the Referee referred to) or by tuning electron filling. In a multiband system like

here, this phase diagram is even richer because $N=5,6,7$ etc. are all commensurate fillings that harbor the Mott insulating state. The corresponding phase diagram, taking into account the crystal-level splittings, was outlined in R. Yu, Zhu and Si, *Current Opinion in Solid State and Materials Science* 17, 65 (2013), which is now cited as Ref. 37; see Fig. 7 of that reference (or Fig. 4 of F. Eilers et al, arXiv:1510.01857, in press at PRL). The key point to emphasize is that, with a crystal level splitting and orbital selectivity, both $N=5$ and 6 can have orbitals at half-filling giving rise to a Mott insulator. This type of theoretical framework goes back to the earlier studies of some of the present theory co-authors at the beginning of the iron-based-superconductor field. The important work of Ishida and Liebsch, Misawa et al. and de 'Medici et al are done in a similar framework, and citing these references are certainly appropriate; we have added them as Refs. 9-11 in the revised manuscript. In our present work, we have theoretically shown that taking into account the Cu-blockage effect in a similar theoretical framework provides the understanding for the origin of the Mott insulating state experimentally discovered here. We feel that this new theoretical understanding is one of the important accomplishments of the present work and, at the same time, falls within the theoretical framework of an overall phase diagram: with one axis labeling (interaction/kinetic energy) tuning, and the other axis labeling electron filling tuning (which, in our case, spans between $N=6$ and $N=5$).” The important point is that, starting from the very first work of Si and Abrahams, proximity to Mott has always been recognized in the multi-band context and involves both bandwidth and doping tunings. Still, the referee is correct, and that Refs. 9-11 deserve to be cited in the same context.

In our previous revision, we have modified the concluding discussion in this spirit. The referee is now satisfied with this revised version of the concluding discussion, stating that “the two points are reasonably well stressed in the conclusions”.

Here, the referee is reminding us that the one sentence in the introduction part should also be rephrased into the same form. We have followed the advice of the referee and revised the sentence in the introduction so that the referencing is done in the same way as in the conclusion, including to Refs. 9-11.

“I also find quite unfair to quote Ref. 37 for the “general theoretical investigation of a Mott insulating state in an overall phase diagram”. The diagram of Ref. 37 is a schematic summary of many previous investigations including very basic physical facts and less obvious aspects related to the Hund’s coupling. I think that the above sentence should quote all the above theoretical works where the possible role of both the $n=6$ and $n=5$ Mott transitions has been discussed, including 9-11.

We again thank the referee for the comment. The reason that Ref. 37 (now Ref. 38) is used to refer to the general theoretical investigation of a Mott insulating state in an overall phase diagram is precisely because it summarized all the earlier works (including Si and Abrahams, Yu and Si etc..) in addition to the more recent studies. As already mentioned, proximity to Mott always means proximity via BOTH bandwidth tuning and doping tuning, and this predated Refs. 9-11. Nonetheless, Refs. 9-11 do represent important contributions in this overall context and they are indeed included in the citations of the same sentence.

I am satisfied by the other replies by the authors.”

We thank the referee once again for the positive comments regarding our revision.

“Finally, a very minor point. When the authors conclude that the role of strong correlations can be unifying framework of several superconductors, they might add also the alkali-doped fullerides, where the role of strong correlations seems nowadays established [See for example, the recent paper by Nomura et al. Science Advances 1, e1500568 (2015)]”

We thank for the referee for suggesting alkali-doped fullerides to be included in the concluding discussion. To conform to the broad nature of this concluding sentence, we have chosen to refer to the first experimental paper demonstrating the Mott insulating state in the overall phase diagram of this class of superconductors (the new Ref. 40).

“In conclusion, I think that the present manuscript can be published in Nature Communications once the authors take into account my last comments.”

We have addressed all the comments of the referee in our revised manuscript. We thank the referee for his/her recommendation for publication of our work in Nature Communications.

Response to referee 2

“I have carefully read the revised manuscript and the authors’ response. I appreciate the authors’ performing new x-ray experiments to determine the valence state of Cu. I agree with the authors’ assessment that their set of experimental measurements provides strong evidence for a Mott insulating state for samples with concentrations near $x=0.5$.”

We thank the referee for appreciating our efforts and agreeing with us that our experiments establish the samples with concentrations near $x=0.5$ to be a Mott insulating state.

“The main issues, however, are about the novelty of this finding and its impact to the understanding of superconductivity in iron-based compounds. On the latter issue, which in my view is the most critical one, the several changes made by the authors are not only unconvincing, but they seem to actually support the opposite point of view, namely that this Mott insulating phase seems nearly inconsequential to the superconducting state. The authors now claim that the Mott insulator near $x=0.5$ is not “near”, but rather “entwined” and “smoothly connected” with the superconducting state at $x=0.016$. To support this claim, they now plot in Fig. 1g the “ordered magnetic moment” as function of x , instead of the Neel transition temperature (which is restricted to the region near $x=0.5$). The first problem is that, since there are no data points for concentrations below $x=0.18$, I do not understand how the authors can justify that the “ordered magnetic moment” line shown in Fig. 1g ends precisely at the end of the superconducting dome at much smaller doping levels, $x=0.04$.”

With regards to the criticism that we have no data for samples below $x=0.18$ and therefore cannot justify the line of ordered moment terminating at the end of the SC dome, we would like to clarify that (1) the main conclusions of our paper do not rely on the line terminating precisely at the edge of SC dome, and (2) the fact that it does is a result of extrapolating the trend seen from the samples we measured. We agree with the referee that it may be misleading to show the line ending precisely at the edge of SC dome and have therefore removed the line for $x < 0.18$.

“The second – and most important – issue is that this plot contradicts Fig. 3c: as the authors state in the main text, there is no long-range magnetic order for $x < 0.44$, as shown by both resistivity and neutron scattering data. Therefore, there is no meaning in assigning an “ordered magnetic moment” in that region of the phase diagram, where at best one has only short-ranged fluctuations with correlation lengths of a few lattice parameters, as shown in Fig. 3c. Finally, as the authors discuss in the paper, the Mott insulating state seems to be favored by the specific Cu concentration of $x=0.5$, which causes real space Fe/Cu order and changes the valence of Fe to $3d^5$. In this case, it is very hard to conceive a scenario in which this peculiar Mott state is “smoothly connected” to the $x=0$ compound.”

The referee criticizes the use of ‘ordered moment’ given that the magnetic order is short ranged. We disagree with the referee because for short-ranged such as glassy magnetic order, the ordered moment is still a well-defined quantity. Indeed, for x sufficiently below 0.5, the system is influenced by disorder, and therefore the spatial order is glassy. Nonetheless, in our DIFFRACTION experiment (which, of course, is a quasi-elastic probe), the measured moment is still (quasi-)static. A scenario of how the real-space Fe/Cu is smoothly connected to a state without Cu is simply that with decreasing Cu concentration, the Fe/Cu ordering becomes more and more short-ranged, which is supported by our data (Supplementary Figure 5). Thus, our experiments show that the Mott state is indeed smoothly connected to the superconducting compound near $x=0$. We also wish to remind the referee that our theoretical calculations presented in the main text and in the supplementary material, provide a natural way to smoothly connect the systems near $x=0.5$ with those near $x=0$, through a trajectory in an overall phase diagram that spans both the bandwidth-tuning and doping-tuning axes.

“On the issue of novelty, although it is true that the STM studies in Ref. [16] could not unambiguously determine the Mott insulating nature of this phase, they did provide solid indirect evidence in favor of this state. Of course, the current work goes well beyond Ref. [16] by extending the phase diagram to $x=0.44$ and by combining a series of thermodynamic measurements that exclude other possible sources for the insulating state. Nevertheless, in my opinion, this does not provide a major advancement, in view of the issues raised above. With appropriate and substantial revisions, the paper might be suitable for publication in a more specialized journal.”

We thank the referee for acknowledging that our work goes well beyond Ref. [16]. As agreed by the other three referees, we believe our work presents several important advances: (1) the elucidation of the origin of the insulating state seen in Ref. [16], especially through the new study in the doping regime close to $x=0.5$, (2) the presence of magnetic ordering which, combining with the transport and other measurements both below and above the Neel temperature, allow an unambiguous identification of the

Mott nature of the insulating state, (3) an understanding from both experimental and theoretical perspectives why such a Mott insulating state is stabilized in $\text{NaFe}_{1-x}\text{Cu}_x\text{As}$, and (4) the experimental evidence for – and the theoretical understanding of – the smooth connection between the Mott insulating state and the superconducting regime.

Response to Reviewer #1:

“The authors have satisfactorily replied to the remaining points that I raised in my report. In particular

1) I appreciate the change in the title, which I think will be beneficial for the impact of the paper.

2) I think that the discussion of the previous literature about the role of Mott physics in iron-based superconductors has been improved.

3) I agree with the authors that the experimental reference on the role of Mott physics in Cs₃C₆₀ is more appropriate than the theory paper I mentioned

Incidentally, I think the authors replied satisfactorily also to the other Referee.

The revised manuscript can be published in Nature Communications in the present form.”

We thank the referee for acknowledging that our revised manuscript satisfactorily addressed the concerns raised in his/her previous report and those raised by the other referees, as well as for recommending the publication of our manuscript in the present form.